# Differentiation of Monofloral Honey Using Volatile Organic Compounds by HS-GCxIMS

**DOI:** 10.3390/molecules27217554

**Published:** 2022-11-04

**Authors:** Hannah Schanzmann, Alexander L. R. M. Augustini, Daniel Sanders, Moritz Dahlheimer, Modestus Wigger, Philipp-Marius Zech, Stefanie Sielemann

**Affiliations:** 1Laboratory of Applied Instrumental Analytical Chemistry, Department Hamm 2, Hamm-Lippstadt University of Applied Sciences, 59063 Hamm, Germany; 2G.A.S. Gesellschaft Für Analytische Sensorsysteme mbH, BioMedizinZentrum, 44227 Dortmund, Germany; 3Dezernat 330 Für Lebensmittel II, Chemisches und Veterinäruntersuchungsamt Ostwestfalen-Lippe, 32758 Detmold, Germany

**Keywords:** GC, IMS, headspace, honey, authenticity, volatile organic compounds

## Abstract

Honey is a natural product and can be described by its botanical origin, determined by the plants from which the bees collect nectar. It significantly influences the taste of honey and is often used as a quality criterion. Unfortunately, this opens up the possibility of food fraud. Currently, various methods are used to check the authenticity of monofloral honey. The laborious, manual melissopalynology is considered an essential tool in the verification process. In this work, the volatile organic compounds obtained from the headspace of honey are used to prove their authenticity. The headspace of 58 honey samples was analyzed using a commercial easy-to-use gas chromatography-coupled ion mobility spectrometer with a headspace sampler (HS-GCxIMS). The honey samples were successfully differentiated by their six different botanical origins using specific markers with principal component analysis in combination with linear discriminant analysis. In addition, 15 honey-typical compounds were identified using measurements of reference compounds. Taking a previously published strategy, retention times of marker compounds were correlated with GC-coupled mass spectrometry (GC-MS) measurements to assist in the identification process.

## 1. Introduction

Honey is a valuable natural product made by bees collecting nectar, honeydew, and pollen from nearby plants. Nectar and honeydew of different plant species vary in terms of, among other things, aromatic substances, mineral content, sugar composition, and water-insoluble components such as pollen. This diversity determines the appearance and flavor of the resulting honey and allows distinguishable characteristics for different botanical origins (called types) [1,2,3,4,5].

Honey may only be designated with the botanical indication if the honey originates wholly or predominantly from the named flowers or plants. In addition, it must have the corresponding organoleptic, physicochemical, and microscopic properties, according to the appropriate Codex Alimentarius published by the Food and Agriculture Organization of the United Nations and the World Health Organization [6].

The color, consistency, taste, and odor are considered in the organoleptic honey analysis. The characteristics of common types of honey are described in the guidelines of the German Food Code. For example, linden honey typically has a color ranging from light yellow (with a green tint) to beige. However, depending on the content of honeydew, it can be much darker. The smell is intensely aromatic, medicinal, and menthol-like with a taste that is intensely aromatic, medicinal, menthol-like, and tart. The consistency is usually crystalline [7].

As part of the chemical-physical analysis, honey is tested explicitly for certain ingredients and parameters that determine the quality and authenticity of the natural product. This can be its sugar contents, electrical conductivity, optical rotation, pH value, or the presence of hydroxymethylfurfural (HMF) [1,3,7]. The results determine various possible aspects of food fraud and safety. This can include incorrect handling [8], an incomplete ripening process [9], or the addition of foreign substances [10], just as well as a false declaration of the botanical or geographical origin.

Honey with a specific botanical origin is also known as monofloral honey and is widely considered more valuable than blends. As a result, there is an increased need for good authenticity testing. In addition to the botanical origin, authenticity testing can include the verification of its geographical origin or the proof of the absence of added extrinsic substances. This work focuses on the botanical origin as this has a direct influence on the volatile organic compounds (VOC) present.

Melissopalynology, the study of the contained pollen using a microscope, is deemed an essential factor for the proof of botanical authenticity. However, this method needs expert operators with a lot of experience and is very time-consuming. Moreover, it is unsuitable for high-throughput analysis [11,12,13]. On the other hand, the advantage of melissopalynology is that the botanical origin and, to some extent, the geographical origin can be checked [11,12,13].

Reliable validation of a honey’s origin needs several unrelated pieces of evidence to make deception more difficult. The previously named chemical-physical parameters can be evaluated in this context. These analyses can be automated and allow for high sample volumes. However, each parameter requires equipment and personnel for the analysis, but multiple characteristics are needed for a conclusive result. Thus, increasing the complexity of such investigations. Unfortunately, current scandals regarding honey show that food fraudsters are targeting these established authentication methods [14,15].

Nuclear magnetic resonance spectroscopy and chemometrics are this field’s most recently established technology [16,17,18,19]. This type of comprehensive analysis is very powerful, as it allows the monitoring of a multitude of attributes. Unfortunately, it is costly in acquisition and operation, so there is an opening for other, more straightforward approaches. One promising avenue is the analysis of the VOC emitted by honey [20].

VOCs of honey were analyzed as early as 1986 concerning its botanical origin [21]. Shortly after, Tan et al. used these compounds to verify the purity of monofloral honey [22]. In other publications, VOCs were used to distinguish honey by geographic [23], botanical [24], or entomological [25] origin. Often, mass spectrometry coupled to gas chromatography (GC-MS) is used in combination with solid-phase-microextraction (SPME) to analyze volatiles [2,26].

Unfortunately, SPME has its drawbacks concerning mechanical robustness and inter-device reproducibility. Other technologies like SPME Arrow or stir bar sorptive extraction try to overcome these difficulties. However, all enrichment processes increase the complexity of the analysis and introduce other disadvantages [27]. An alternative is using a more sensitive detector that makes enrichment obsolete. We used an ion mobility spectrometer (IMS) for this task.

To establish a readily available and rapid screening, we built on the excellent work of Gerhardt et al. (2018) [20]. They used the unique coupling of a standard thermal gradient GC system and an IMS as a new detector technology to establish a temperature gradient-based method. Using samples from three different botanical sources, they showed that using simple, untargeted VOC profiling, authentication of the botanical origin of honey is possible. The utilized GCxIMS system has some known advantages. On the other hand, it requires a certain amount of knowledge to set up and operate properly. To overcome this, we used an easy-to-use system based on a static headspace sampler, an isothermal GC, and a drift tube IMS (HS-GCxIMS) to replicate and expand the previously published results, using a more significant number of samples and a user-friendly software tool for statistical data analysis.

Applying IMS technology in combination with gas chromatographic separation in quality and product control is a growing market. The rapidly increasing number of publications in this field shows the potential of this technique. About ten years ago, the first profound methods focused on olive oil classification [28]. Today a wide variety of foodstuff applications have been published [29,30,31,32]. This includes two other publications targeting the adulteration of honey with syrup [33,34]. Particular advantages are the low detection limits, especially for aroma-active substances in the lower ppb_v_ range, and the possibility of separating and identifying the individual analytes based on their specific drift times. This allows non-targeted analysis with the aid of suitable evaluation algorithms and identifying and quantifying substances as so-called marker compounds for a particular quality criterion [35].

The feasibility study presented here analyzed 58 honey samples from six different botanical origins with a HS-GCxIMS. To analyze the generated complex data, appropriate statistical tools for data reduction and classification were applied. Additionally, implementing and expanding on a previously published strategy [36], the results from the isothermal easy-to-use HS-GCxIMS were correlated with a thermal-gradient HS-GC-MS using retention indices to simplify the identification of substances. Previously this was only done for a thermal-gradient system using multiple flow lines.

## 2. Results and Discussion

### 2.1. Classification of Honey Types Using the Volatile Organic Profile

Honey of different varieties (called types) can be distinguished by their spectrum of emitted volatile organic compounds, as shown in Figure 1. The GCxIMS chromatograms of linden, rapeseed, and chestnut honey demonstrate visually significant differences. Linden honey, for example, offers some volatile compounds with high retention times (>1500 s, Figure 1 at the top), while the others do not. However, VOCs are also found in the lower retention time range, which occurs in only one of the three kinds of honey shown. Looking at the peaks, eluting at a retention time range from 550 s to 700 s, it can be seen that different and more intense peaks appear for linden honey (Figure 1 at the bottom). It is also apparent that the substances found differ in intensity between the honey types as well. Accordingly, a distinction can be made based on intensity in addition to the presence of specific volatile compounds.

All peaks were identified from the 58 GCxIMS measurements and manually labeled using VOCal software (VOCal 0.1.4., G.A.S., Dortmund, Germany). A total of 514 different peaks were selected. Figure 2 shows an example of 23 peaks from the GCxIMS plots of linden, acacia, rapeseed, silver fir, chestnut, and manuka honey placed in a so-called gallery plot. A gallery plot shows the chosen coverage area of the selected peaks within GCxIMS plots and allows the comparison of their intensities in extensive datasets more easily. For example, markers 1, 3, 15, and 21 are only found in manuka honey. Even from the small selection of signals shown in Figure 2, it is clear that each type of honey has at least a few peaks associated with only one of them.

All measurements and the 514 marked peaks were entered into the VOCal software for automatic comparison. Here, labels of the honey types were assigned to the 58 measurements. Next, box plots were used to graphically display the distribution of intensities of markers for the different honey types. This allows the selection of type-specific peaks or those that show a significantly different intensity.

Figure 3 displays examples of box plots of two unidentified marker compounds (markers 1 and 2). On the x-axis, the honeys are grouped by type, while the y-axis represents the intensities of the marker compound found in the samples. For example, marker 1 occurs exclusively in manuka samples at an intensity greater than 0.37 V. Furthermore, marker 2 is only found in linden honey at an intensity greater than 0.42 V.

As shown at the example from Figure 3, all 514 markers were tested in this way for the potential to distinguish honey types. This selection process includes markers that are specific for a single honey type, either by presence or by absence, just as well as markers that allow differentiation between two or three types. A total of 113 markers were selected that have the potential to distinguish honey types from each other. The majority of markers (38 and 21) were found for linden and manuka honeys. Fewer markers were identified for the other honey. These were in each case: acacia (3), rapeseed (4), silver fir (1) and chestnut (3). The remaining 43 markers were selected because they differed in at least three of the classes.

The results obtained show that a distinction can be made based on one or more specific markers. Still, the spectra contain other peaks of relevant substances in a particular ratio. Only in this case, a simple observation of the individual intensities is no longer sufficient, and statistical methods are needed to compare the results with each other. Therefore, a principal component analysis (PCA) is used.

In the analysis presented in the next section, intensity data of marker compounds (GCxIMS measurement) were initially preprocessed by means of the PCA algorithm for the reduction in the dimensions of the initial dataset, since most of the initial marker compound information carried by the intensities can be maintained in some principal components (PCs) only. Then, the preprocessed data were introduced into the linear discriminant analysis (LDA) algorithms for the construction of the corresponding predictive visualization (only marker compounds with at least 5% variance).

For PCA-LDA, the intensities of the 113 selected markers were used as a dataset. Therefore, the data matrix had a dimension of 58 (samples) × 113 (markers). The PCA-LDA visualization shows a clear separation between each group of samples (Figure 4).

The presented classification method for honey was tested with two honey samples in an out-of-sample forecast. One was purchased from a local store (linden), and the other was provided as linden honey, but with the remark of being “unauthentic”. Using the PCA-LDA, the sample from the local store was correctly classified as linden-type (Figure 5B). The non-authentic sample, originally declared as linden honey, was identified as rapeseed honey. This identification was supported by external tests. The out-of-sample forecast thus shows that the PCA-LDA has the potential to identify the correct type among the six types by means of VOCs from the HS.

The described approach allows a specific separation of similar honey types. This is especially important for honey of the honeydew kind, as honeydew honey is tough to assign a botanical origin through pollen analysis. The pollen analysis is not directly applicable to honey types that do not originate from flowers. The differentiation of honeydew honey using the volatile profile could solve this problem [37]. Our research corroborates this possibility.

### 2.2. Identification of Substances

In addition to classifying honey according to its botanical origin using statistical methods, the question arises as to which analytes are specifically involved in the gas space. Since the individual VOCs in the GCxIMS plot have characteristic drift and retention times, it is possible to identify them. Since there is hardly any database for IMS spectra, the assignment is carried out through parallel mass spectrometry. In doing so, fifteen marker compounds from GCxIMS measurements were identified using the recently published correlation strategy based on HS-GC-MS measurements. These were verified by comparing signals’ retention and drift times from analyzed reference substances. Retention times and reduced mobilities are listed in Table 1. An exemplary GCxIMS-plot with the marked signals is shown in Figure 6.

As seen in the GCxIMS plot in Figure 6, only 15 compounds were safely identified. There are still numerous peaks in the plot that could not be assigned to a substance with sufficient validity in the course of this feasibility study. A comparison with the literature confirms these initial results. In particular the aldehydes from pentanal to decanal, benzaldehyde, and furfural, as well as alkanones such as 2-butanone and 2-pentantone along with terpenes such as α-pinene, were frequently identified in the headspace of different honey types [2,38].

Thus, these substances are individually not characteristic of a single honey type but only allow a conclusion about the authenticity in combination. Nevertheless, these first assignments served to show the potential of the GCxIMS and the ability to break away from the usual non-target analytics. Therefore, a database for VOCs emitted by honey will be set up in a further research project, allowing the valid identification of more substances. In this way, it will be possible to compare the GCxIMS results with the results of analytical techniques with higher resolution separation and more complex methods such as GC-MS. This method can tentatively identify compounds and has been shown to be able to identify more substances such as 2-acetyl-l-pyrroline, methional, phenylacetaldehyde, 1-hexen-3-one, 2-phenylethanol, p-cresol, p-anisaldehyde, eugenol, and vanillin as markers for linden, but also acacia honey [39]. Ideally, all markers characteristic of a honey type can be assigned to a substance so that a better understanding of the cause of distinguishability can be achieved in classification.

## 3. Materials and Methods

### 3.1. Chemicals and Samples

Reference substances were purchased from Fisher Scientific (Schwerte, Germany), Carl Roth (Karlsruhe, Germany), TCI Deutschland (Eschborn, Germany), and Sigma-Aldrich (Taufkirchen, Germany) in sufficient purity (90% or more). A saturated solution of sodium chloride was prepared using NaCl (purity ≥ 99.5%) from AppliChem (Darmstadt, Germany) and ultra-pure water, produced with a Veolia Elga Purelab flex 4 (Celle, Germany). The in-house generator supplied the nitrogen gas with a purity of at least 99.999%.

Retail honey samples with specifically named botanical origins were provided by the Chemisches und Veterinäruntersuchungsamt Ostwestfalen-Lippe. These samples had been analyzed in line with public monitoring using state-of-the-art methods before being made available to us as authentic samples.

Samples were authenticated based on their electrical conductivity, water content, HMF content, color, pollen content, sugar profile, non-target ^1^H-NMR, and organoleptic profile. A summary of the samples’ botanical origin is given in Table 2. Their geographical origin is predominantly from Germany and other blends originating in EU and non-EU countries, except for the manuka honey, which is a product of New Zealand. Additionally, two kinds of honey for out-of-sample forecast were provided: one linden honey from Germany was purchased from a local store, and the other honey was provided to us as an unauthentic linden honey of undisclosed origin.

The samples were stirred thoroughly before being taken from their original container. As sample preparation, 2 g (±0.02 g) honey and 2 mL saturated NaCl solution were filled into a 20 mL vial and crimp-closed with a PTFE/Polysiloxane septum (CS—Chromatographie Service, Langerwehe, Germany). Samples were prepared in triplicate. For blank samples, 2 mL of the NaCl solution was analyzed. Reference substances were diluted in ultra-pure water to 100 μg/L. Finally, 1 mL of these solutions was filled into a vial for the analysis.

### 3.2. Instrumentation

The analysis was done on an easy-to-use FlavourSpec (G.A.S., Dortmund, Germany) HS-GCxIMS. Samples were automatically shaken for 30 min at 80 °C. 1 mL of headspace was injected into the injector at 80 °C without split flow. Substances were separated on a 15 m × 0.53 mm × 1 μm MXT-5 column (Restek, Bellefonte, PA, USA) with a nitrogen carrier gas ramp (2 mL/min for 1 min., in 14 min. up to 150 mL/min, this held for 35 min, for a total of 50 min. for the analysis). Detection was carried out with a drift tube of 15.2 mm (diameter) and 98 mm (length) ionization by ^3^H-source, at 45 °C, a nitrogen drift gas flow rate of 150 mL/min, and a field strength of 500 V/cm.

Reference measurements were run on a thermal-gradient HS-GC-qMS (Shimadzu, Kyoto, Japan). Using the same vials, samples were incubated for 30 min at 120 °C, afterwards injecting 2 mL of the headspace at a split of 1:10. The separation was performed on a HP-5 MS UI (Agilent, Santa Clara, CA, USA) 30 m × 0.25 mm × 0.5 μm column at a constant flow rate of 35 cm/s helium and a temperature ramp (50 °C held for 4 min, at 5 °C/min to 150 °C, at 10 °C/min to 200 °C). Analytes were ionized using electron-impact ionization at 70 eV and 200 °C with a scan range of 32–300 *m*/*z* at an event time of 300 ms.

### 3.3. Data Analysis

The GCxIMS-plots were interpreted manually, using the G.A.S. software VOCal 0.1.4. Peaks were selected manually, and their position was determined by choosing a coverage area. This area consists of the retention (in seconds) and drift time (in milliseconds) range in which the selected peak is situated and then added to the data set for further use. The signal intensity (in V) was determined by the software automatically using the maximum of the selected area.

For data processing, an initial step of peak alignment was carried out. A drift time scale using a reactant ion peak (RIP) as reference was carried out automatically by the software. Then, peaks (to be called markers) were selected by visual exploration of the GCxIMS plots of each sample and their intensities were selected as the analytical signals.

A non-supervised principal component analysis using auto-scales was carried out in order to reduce the dimensionality. Then, PCA scores were taken to carry out a linear discriminant analysis. The LDA incorporated PCA components automatically, carrying out at least 5% of variance.

### 3.4. Identification of Substances

Selected peaks in the GCxIMS measurements were identified using the previously published correlation strategy [36]. Signals from measurements acquired with the Flavourspec GCxIMS were correlated with the signals from corresponding measurements with a HS-GC-MS using retention indices. The mass spectrum of those selected peaks and the MS database (NIST/EPA/NIH Mass Spectral Library 14 from the National Institute of Standards and Technology of the U.S. Department of Commerce) allowed a tentative identification. As a follow-up, the corresponding reference substances were analyzed on the HS-GCxIMS for confirmation by comparing drift and retention times.

## 4. Conclusions

Using a commercially available isothermal easy-to-use HS-GCxIMS, we could establish an analytical method to identify the botanical origin of store-bought honey samples based on the profile of VOC. By selecting a series of essential signals and running a PCA-LDA, we could differentiate between six of the most important honey types available in Germany by their botanical origins (linden, manuka, acacia, rapeseed, chestnut, and silver fir). This was verified using an out-of-sample forecast with the basis of the previous proven authentic samples. It should be noted that when using such statistical tools, larger sample sizes per honey type provide more accurate information. Random honey from a local store and an unauthentic sample were analyzed. Both were placed in the correct category based on the classification method. Organoleptic and other external tests supported these findings. The use of the profile of VOC adds another authentication method for honey samples to the toolbox of the authorities and private laboratories. However, the wide variety of substances monitored using this method increases the efforts and costs required to deceive inspectors.

In addition to the classification using unidentified markers, 15 substances present in the headspace of the honey samples were identified using the measurement of reference standards and matching the retention and drift time. The previously published correlation strategy allowed a faster and simpler screening for relevant substances. Identified marker substances can be validated with other publications, in addition to the possible identification of volatile contaminants or byproducts [1]. The ease of use and the ability to identify trace compounds enables this method to be used alongside the commercially introduced non-targeted NMR analysis, which focuses on higher concentrated compounds.

By expanding on the previous work of Gerhardt et al. (2018) [20] using a powerful benchtop system, transferring the method onto a more straightforward and readily available easy-to-use HS-GCxIMS as a further step toward routine application, it is now possible to enlarge the set of samples and honey types to cover all available products, incorrect processing, or storage conditions.

## Figures and Tables

**Figure 1 molecules-27-07554-f001:**
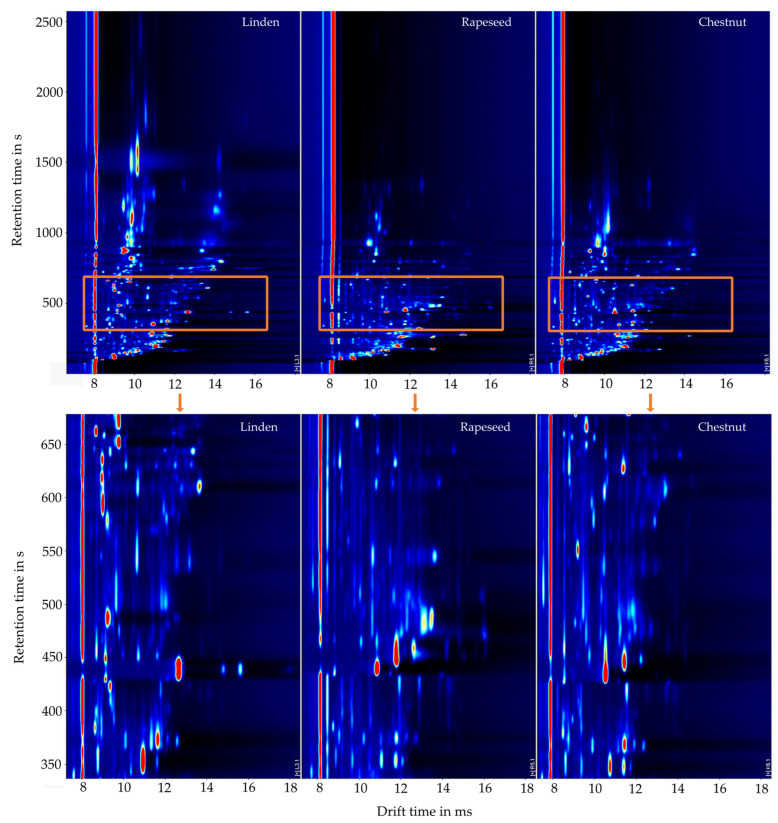
Gas chromatography ion mobility spectrometry (GCxIMS) plot of three honey types (from left to right: linden, rapeseed, and chestnut) in the upper part. In the lower part, the retention time range from 340 ms to 700 ms (orange rectangular) is enlarged.

**Figure 2 molecules-27-07554-f002:**
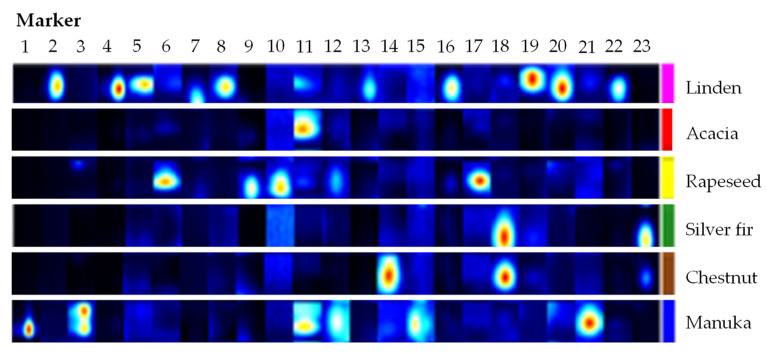
Gallery plot of 23 marker signals for linden, acacia, rapeseed, silver fir, chestnut and manuka type honey.

**Figure 3 molecules-27-07554-f003:**
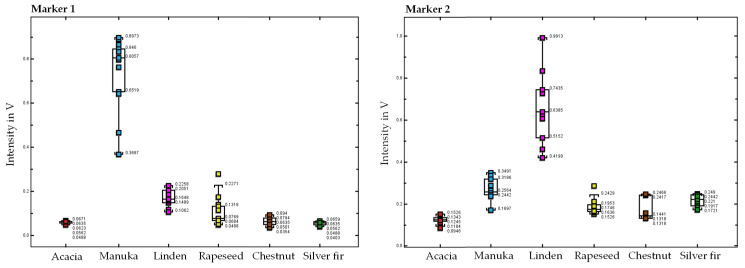
Box plots for two exemplary marker compounds (**left marker 1** and **right marker 2**), with the intensity on the y-axis and the samples grouped by honey type on the x-axis (acacia, manuka, linden, rapeseed, chestnut, and silver fir).

**Figure 4 molecules-27-07554-f004:**
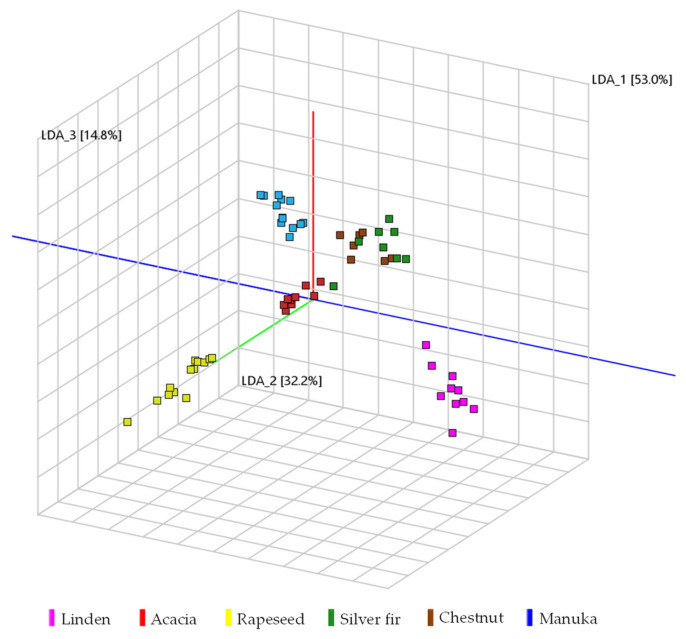
Three-dimensional visualization of principal component analysis with linear discriminant analysis (PCA-LDA) of the honey data set with samples grouped by color as honey type (linden = pink, acacia = red, rapeseed = yellow, silver fir = green, chestnut = brown, and manuka = blue).

**Figure 5 molecules-27-07554-f005:**
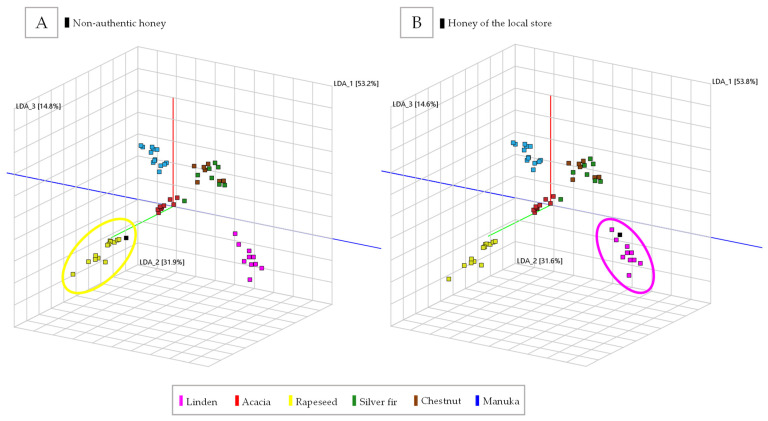
Three-dimensional visualization of the PCA-LDA of the honey data set with samples grouped by color as honey type (linden = pink, acacia = red, rapeseed = yellow, silver fir = green, chestnut = brown, and manuka = blue). (**A**) shows the PCA-LDA with the non-authentic honey, which can be classified as rapeseed honey (yellow circle). (**B**) displays the classification of the honey from the local store, which can be classified as linden honey (pink circle).

**Figure 6 molecules-27-07554-f006:**
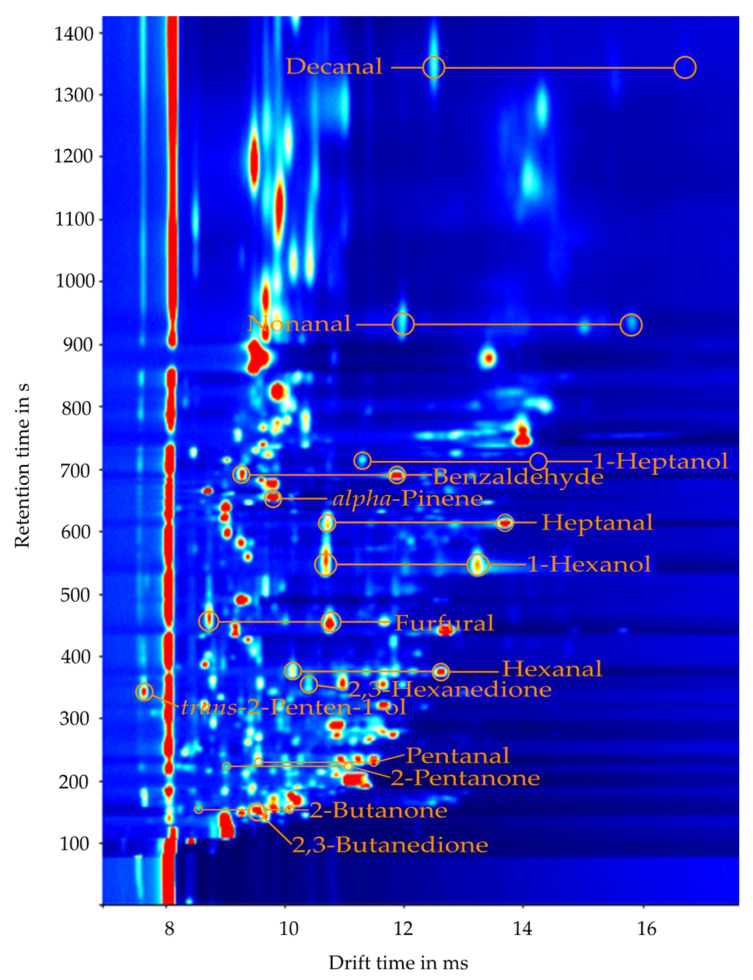
Partial GCxIMS plot of a store-bought linden-type honey with the identified monomer and dimer signals labeled.

**Table 1 molecules-27-07554-t001:** Retention times and reduced ion mobility (K_0_) for the monomers and dimers of the identified substances, including the CAS numbers.

Substance	CAS Number	Retention Time in s	K_0_^Monomer^ in cm^2^(Vs)^−1^	K_0_^Dimer^ in cm^2^(Vs)^−1^
2,3-Butanedione	431-03-8	151	1.77	
2-Butanone	78-93-3	153	1.97	1.67
2-Pentanone	107-87-9	220	1.86	1.52
Pentanal	110-62-3	232	1.77	1.46
*trans*-2-Penten-1-ol	1576-96-1	341	2.19	1.29
2,3-Hexanedione	3848-24-6	355	1.62	
Hexanal	66-25-1	373	1.65	1.34
Furfural	98-01-1	446	1.91	1.57
1-Hexanol	111-27-3	546	1.57	1.27
Heptanal	111-71-7	611	1.57	1.23
*α*-Pinene	80-56-8	653	1.71	
Benzaldehyde	100-52-7	691	1.81	1.42
1-Heptanol	11-70-6	713	1.49	1.18
Nonanal	124-19-6	942	1.41	1.06
Decanal	112-31-2	1360	1.35	1.01

**Table 2 molecules-27-07554-t002:** Summary of the analyzed honey samples, differentiated by their type of botanical origin.

Honey Type	Botanical Origin	Number of Samples	Color Coding
Acacia	*Robinia pseudoacacia*	10	Red
Chestnut	*Castanea sativa*	7	Brown
Linden	*Tilia* spp.	11	Magenta
Manuka	*Leptospermum scoparium*	9	Blue
Rapeseed	*Brassica napus*	13	Yellow
Silver fir	*Abies alba*	8	Green

## Data Availability

The datasets generated for this study are available on request to the corresponding author.

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
