# Peer review of "Differentiation of Monofloral Honey Using Volatile Organic Compounds by HS-GCxIMS"

_molecules, 2022, doi:10.3390/molecules27217554_

Round 1

Reviewer 1 Report

Headspace Gas Chromatography is a sampling and/or injection technique involving the indirect determination of volatile constituents in liquid or solid samples by analysing the associated vapour phase. In this manuscript, the mentioned method was applied in order to identify the plants whose nectar was used to make honey. Because of consumers, their habits and rules in declaring and controlling the honey trade, it is of great importance to determine the authenticity of honey that is declared as honey obtained from one type of plant (monofloral honey).

The introduction is well written and provides sufficient information about the motivations for setting up this essay and the state of the art in the field.

Line 83-88 - they do not belong in the introduction. Delete or move to the appropriate chapter

Results - Results are presented clearly and accurately. The author guides us from the visualization of results during the application of this method, to the visualization of numerical values, and finally, using PCA analysis, he confirms the possibilities of correct classification of honey using the mentioned method. When identifying the substances, the authors used the reference material, but also made certain modifications, so that the identification would be more accurate and precise.

MandM - MandM are well described and allow repeatability of experiments. However, in the material and methods it is necessary to additionally describe how the honey was sampled. Provide a precise trade declaration of honey and geographical origin. It is not enough to state that these are retail samples of honey.

Discussion - The discussion is the weakest part of this manuscript. In the discussion, the authors were based only on the methodological aspects of this work. However, the discussion should be expanded from the aspect of the quality and composition of honey and from the aspect of certain properties of substances that were discovered by the method used in this work. This would contribute to the reader understanding why the methods used and the substances detected are of particular importance in the process of quality analysis and control of the origin of honey.

The references are well chosen, and you should double check that they are all adapted to the MDPI style.

Author Response

Dear Reviewer,

thank you for your review and support of our submission, "Differentiation of Monofloral Honey using Volatile Organic Compounds by HS-GCxIMS". We have taken your ideas into account and modified the article accordingly.

The honey was provided to us as store-bought samples. Unfortunately, this means the sample information we have is limited to the information provided by the distributor. The usual geographical origin is "from countries inside and outside of the European Union," and the botanical origin is the type we named in the article. The named plant species is based on what is allowed to be used for this type of honey. Usually, retail honey is a blend to achieve a fair quality product that the consumer expects. Only manuka honey is described as a product of New Zealand, and some honeys are declared to be a product of German origin. We added this information in lines 258-259. We know that using less blended honey and a larger sample size could allow the investigation of differences in geographical origins. Unfortunately, this is not possible with this set of samples. However, we are currently hoping for a larger project with a high volume of non-retail samples with more detailed information on their origin.

We changed lines 83-88 (now line 109 ff) as we agree that they don’t belong to the introduction.

Thank you, especially for your advice on our discussion. Due in part to another reviewer's comments, we have renamed the results section to "Results and Discussion" and included a "Conclusion" section to add a closing summary. We hope this meets your expectations.

We appreciate your input and valuable effort.

With best regards

The authors

Reviewer 2 Report

Dear,

I have attached my comment as following.

Regards,

Akira Wanikawa

Author Response

Dear Mr. Wanikawa,

thank you very much for your review and support of our submission, "Differentiation of Monofloral Honey using Volatile Organic Compounds by HS-GCxIMS". We have taken your ideas into account and modified the article accordingly.

Mainly we changed the introduction as recommended. We explained the standard methods for the quality control for honey in more detail in lines 54ff and briefly discussed the advantages and disadvantages. The benefit of the used IMS technology is explained in lines 98-108, where we additionally give examples for applications where else the IMS method is used nowadays. 

We also added information about the organoleptic, physicochemical, and microscopic properties and how they were measured. They are described in the guidelines of the German Food Code. We used linden honey to give more general information as an example. We fully agree that this makes the paper more interesting for people outside the honey application field. However, we are unable to provide the results of the organoleptic, physicochemical and microscopic tests, which were done on the analyzed honey in our study. These analyses were done by an accredited German state laboratory (Chemisches und Veterinäruntersuchungsamt Ostwestfalen-Lippe, Detmold, Germany). Their results are confidential and may not be published. We are unable to redo these analyses in our labs to recreate the results, as we lack specific equipment and experience for the organoleptic tests.

Thanks also for the advice concerning the discussion. As you suggested, we have renamed the results section to Results and Discussion and tried to give some more details. Especially we further deepen the discussion about the identified compounds. On your suggestion, we also added the "Conclusion" as a closing summary. We hope this meets your expectations.

With best regards

The authors

Reviewer 3 Report

Review Report

Manuscript ID: molecules-1968933

Title: Differentiation of Monofloral Honey using Volatile Organic Compounds by HS-GCxIMS

Journal: Molecules

In this study, isothermal HS-GCxIMS was used to establish an analytical method for the identification  of the botanical origin of store-bought 58 honey samples. Authors have shown that by careful selection of  a series of essential signals and running a PCA-LDA, it is possible to differentiate six of the most important honey types (linden, manuka, acacia, rapeseed, chestnut, and silver fir) by their botanical origins,  which are obtainable in Germany.

This study has the potential to be cited. 

I recommend that the Editorial Office consider this manuscript for publication in the present form.

Author Response

Dear Reviewer 3,

thank you for your review and support of our submission "Differentiation of Monofloral Honey using Volatile Organic Compounds by HS-GCxIMS".

With best regards

The authors